# USING GANS FOR GENERATION OF REALISTIC CITY-SCALE RIDE SHARING/HAILING DATA SETS

## ABSTRACT

This paper focuses on the synthetic generation of human mobility data in urban areas. We present a novel and scalable application of Generative Adversarial Networks (GANs) for modeling and generating human mobility data. We leverage actual ride requests from ride sharing/hailing services from four major cities in the US to train our GANs model. Our model captures the spatial and temporal variability of the ride-request patterns observed for all four cities on any typical day and over any typical week. Previous works have succinctly characterized the spatial and temporal properties of human mobility data sets using the *fractal dimensionality* and the *densification power law*, respectively, which we utilize to validate our GANs-generated synthetic data sets. Such synthetic data sets can avoid privacy concerns and be extremely useful for researchers and policy makers on urban mobility and intelligent transportation systems.

## 1 INTRODUCTION

Ride sharing or hailing services have disrupted urban transportation in hundreds of cities around the globe (Times, 2018; Cohen & Kietzmann, 2014). In United States, it has been estimated that between $24\%$ to $43\%$ of the population have used ride-sharing services in 2018 (Recode, 2018a). Uber alone operates in more than $600$ cities around the globe (Recode, 2018b). Ride sharing services have turned urban transportation into a convenient utility (available any place at any time), and become an important part of the economy in large urban areas (Hahn & Metcalfe, 2017).

Ride request data from ride sharing services can potentially be of great value. Data gathered from ride sharing services could be used to provide insights about traffic and human mobility patterns which are essential for intelligent transportation systems. Ride requests in major cities with high penetration by such services exhibit spatial and temporal variability. Modeling of such variability is a challenging problem for researchers. Moreover, there are still unresolved challenges, such as: optimal algorithms for dynamic pooling of ride requests (Chen et al., 2017), real-time pre-placement of vehicles (Jauhri et al., 2017b), and city scale traffic congestion prediction (Maystre & Grossglauser, 2016) and avoidance. Access to large amount of actual ride request data is essential to understanding and addressing these challenges.

Data from ride sharing services have been used for real-time sensing and analytics to yield insights on human mobility patterns (Song et al., 2010; Jauhri et al., 2017a). Each city exhibits a different pattern of urban mobility – there could be cultural or economical factors governing these patterns. If ride sharing services constitute a significant percentage of the riders in a city, can we build models from ride request data to model urban mobility for the whole city and provide societal benefit without compromising personal privacy? This question motivates us to explore the potential of using Generative Adversarial Networks (GANs) to generate synthetic ride request data sets that exhibit very similar attributes as the actual ride request data sets.

This work proposes a novel approach of generating synthetic ride request data sets using GANs. This approach involves viewing ride requests as a (temporal) sequence of (spatial) images of ride request locations. The approach uses GANs to match the properties of the synthetic data sets with that of real ride request data sets. Many recent works using neural networks have looked at demand prediction (Yao et al., 2018b; Zhou et al., 2018) and traffic prediction at intersections (Yao et al., 2018a). In our work, we are looking at generating actual ride requests for both spatially and temporally granular intervals. Also, we compare and validate the spatial and temporal variations of the

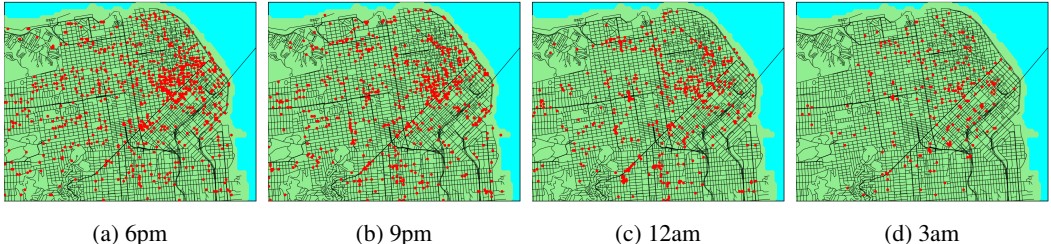

|              |              |              |              |
|:------------:|:------------:|:------------:|:------------:|
| (a) 6pm      | (b) 9pm      | (c) 12am     | (d) 3am      |

Figure 1: Ride requests for a small region of downtown San Francisco for a typical week day. Each figure shows the aggregated ride-locations (red dots) over a period of an hour. Each red dot may represent one or more ride-locations. Ride density varies spatially and temporally.

synthetic data sets with the real data sets. In dealing with large amount of data for many cities and long training times for GANs, we develop effective ways to parallelize and scale our GANs training runs using large CPU clusters on AWS. We present our GANs scaling approach and experimental results, and show that significant reduction in training times can be achieved.

## 2 DATA SETS FROM RIDE SHARING SERVICES

In this section, we introduce the actual (real) ride request data sets used for our GANs training and evaluation. We use the real data sets to compare with and validate the GANs generated synthetic data sets.

Our real ride request data sets consist of all the ride requests for an entire week for the four cities. There is a strong repeating pattern from week to week as shown in Figure 2. Hence the week-long data should be quite representative. For all four cities, the ride sharing services have significant penetration. Hence we believe the ride request data sets also reflect the overall urban mobility patterns for these cities.

### 2.1 RIDE REQUESTS

Our real data sets are actual ride requests for four cities over one week period from ride sharing services operating in the United States. Each ride request in the data set includes: request time and pickup location (latitude & longitude), and drop-off time and location (latitude & longitude). For this work we focus on ride request time and pickup location for generating pickup locations; and ride request time and drop-off location to generate drop-off locations. After training independent GANs models for pickup and drop-off locations, we generate synthetic locations using GANs and leverage graph generator approach (Easley & Kleinberg, 2010; Jauhri et al., 2017a) to pair all pickup and drop-off locations to obtain synthetic ride requests. The trajectory or optimal route for a ride is not within the scope of this work.

For the rest of the paper, we will use the term *ride-locations* to refer to both pickup and drop-off locations wherever they can be used interchangeably.

We do temporal and spatial quantization of the raw ride request data from ride sharing services. We partition the entire week into 2016 time intervals of 5 minutes each, and lump together all the ride requests within each interval. We partition spatially the area of the entire city into small squares with side length, $\epsilon$, of 50 meters, and lump together all the ride-locations occurring within the same square area. Each square area is then represented by a single pixel in a 2-D image with the gray scale intensity of the pixel reflecting the number of ride-locations in that square area (in a time interval). Occurrence of no ride-locations in an area is denoted by zero pixel intensity; positive integers $(1, 2, 3, \ldots)$ as pixel intensity denote the number of ride-locations in the square area.

Combining the temporal and spatial quantizations, the real ride request data set for each city becomes a time sequence of images with each image spatially capturing all the ride requests occurring in a particular 5-min interval.

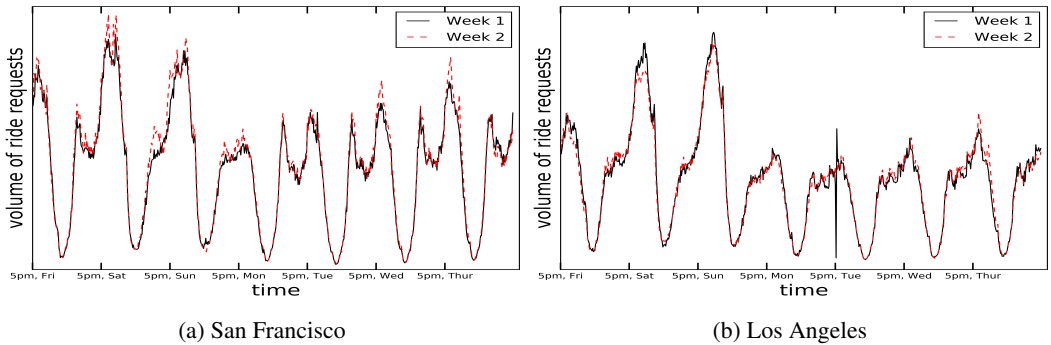

(a) San Francisco  (b) Los Angeles

Figure 2: Weekly ridership (y-axis) pattern of ride requests.

## 2.2 SPATIAL AND TEMPORAL PATTERNS

The actual ride requests in every city exhibit distinct patterns of variability in both the spatial dimension (over geographical area of the city) and the temporal dimension (over each day and over each week). In Figure 1, this variability is illustrated. The ride request density is at its highest at 6pm, and continually decreases over time till 3am. Spatially there are dense patches of ride requests and these dense patches can shift with time, reflecting shifting concentrations of commuters in different areas at different times of day. We observe similar repeating patterns of temporal and spatial variability for all four cities.

Previous works have been able to characterize these temporal and spatial variability patterns (Jauhri et al., 2017a). A graph can be used to model the ride requests within a 5-min interval, with nodes [1] representing pickup and drop off locations and a directed edge connecting the pickup node and the drop-off node. It was shown in (Jauhri et al., 2017a) that the size and density of this Ride Request Graph (RRG) evolves in time in response to the fluctuation of ride requests during each day and through out each week.

It was observed that these ride request graphs exhibit and obey the *Densification Power Law* (DPL) property, similar to other graphs modeling human behaviors such as social networking graphs and publication citation graphs (Leskovec et al., 2005). It was further observed that the ride request graphs for each city exhibit a distinct degree or *exponent* of the DPL, and that this *DPL Exponent* ($\alpha$) can be viewed as a very succinct quantitative characterization of the temporal variability of the ride request patterns for that city. For any time snapshot $t$:

$$e(t) \propto n(t)^{\alpha} \tag{1}$$

where $e(t)$ and $n(t)$ are the number of edges and number of nodes respectively, formed by all ride requests occurring in the time interval $t$. Edge weight denote the number of requests from the same source (pickup) to destination (drop-off) nodes in time snapshot $t$. The number of edges grows according to a specific exponential power ($\alpha$) of the number of nodes.

There is also a comparable quantitative characterization of the spatial variability of the ride request patterns for each city. The actual geographical locations of the nodes of the ride request graphs is not explicitly represented and therefore another characterization is needed. *Correlation Fractal Dimension* (Schroeder, 2009; Belussi & Faloutsos, 1998) provides a succinct description of a k-dimensional point-set to provide statistics about the distribution of points; it provides a quantitative measure of self-similarity. The spatial distribution of ride requests in each time interval can be viewed as a point-set image. We can measure the Correlation Fractal Dimension ($D_2$) as described in (Jauhri et al., 2017b). Values for correlation fractal dimension computed for each time snapshot $t$ fall within a range for each city indicating the degree of self-similarity, and the consistent weekly pattern. For our 2-dimenional space, we impose a 2D-grid with square of side $\epsilon$ [2]. For the i-th square, let $C_{\epsilon,i}$ be the count of requests in each square. The correlation fractal dimension is defined

---

[1] Each node covers a geographical region of a square of length $\epsilon$. Nodes within which no ride requests occur are not considered in the graph.

[2] $\epsilon$ parameter is equivalent to the square length for representing a node in the Ride Request Graph.

as:

$$D_2 \equiv \frac{\partial \log \sum_i C^2_{\epsilon,i}}{\partial \log \epsilon} = \text{constant} \qquad \epsilon \in (\epsilon_1, \epsilon_2) \tag{2}$$

For self-similar data sets, we expect the derivative to be constant for a certain range of $\epsilon$ (Traina Jr et al., 2010). We observe that this range varies for our four cities, and each city exhibits a distinct value range for its correlation fractal dimension ($D_2$).

We use the *Densification Power Law Exponent* ($\alpha$) and the *Correlation Fractal Dimension* ($D_2$) to capture and characterize the temporal and spatial variability, respectively, for the ride request patterns for each city. RRG created for every time snapshot captures ridership fluctuations over time; nodes in a RRG do not encode any spatial information. Therefore, we compute Correlation Fractal Dimension for each time snapshot to capture the spatial distribution of both pickup and drop-off locations. The temporal evolution, and spatial distribution at any give time snapshot capture the dynamics of ride requests. We use these two parameters independently to confirm the similarity between the real data sets and the GANs generated synthetic data sets. We can claim strong similarity if the values of these two parameters ($\alpha$ and $D_2$) of the synthetic data sets match closely the values of the same two parameters of the real data sets.

## 3 GENERATING RIDE REQUESTS USING GANS

### 3.1 IMAGE GENERATING USING GANS

Generative Adversarial Networks learn to generate high quality samples (Goodfellow et al., 2016) i.e. sample from the data distribution $p(x)$. Previous works by (Denton et al., 2015; Ledig et al., 2017) synthesized images of a higher quality using GANs which were hard for humans to distinguish from real images. Conditional GANs are an extension of GANs to sample from a conditional distribution given each image has an associated label which is true for our case of ride requests.

In our framework, we would apply conditional GANs using ride request data in the form of images; similar to as shown in Figure 1 but without the base map shown in color.

### 3.2 USING GANS FOR RIDE REQUEST GENERATION

GANs learn a mapping from a random noise vector $z$ to output image $x$. Conditional GANs learn a mapping from noise vector $z$ and a label $y$ to $x$ (Mirza & Osindero, 2014; Gauthier). The additional variable in the model allows to generate and discriminate samples conditioned on $y$. The generator accepts noise data $z$ along with $y$ to produce an image. The discriminator accepts an image $x$ and condition $y$ to predict the probability under condition $y$ that $x$ came from the empirical data distribution rather than from the generative model. The objective function can be expressed as:

$$\mathbb{E}_{x,y \sim p_{data}(x,y)}[log D(x,y)] + \mathbb{E}_{z \sim p(z), y \sim p_y}[log(1 - D(G(z,y),y))] \tag{3}$$

where $G$ tries to minimize to this objective function against an adversarial $D$ that tries to maximize it.

### 3.3 TRAINING PROCESS & ARCHITECTURE

Every image is assigned a label from the set $\{0, 1, 2, ..., 23\}$ representing the hour of a day. All twelve instances of five minute snapshots within an hour are assigned the same hour label [3]. To accelerate our training using multiple machines, we exploit spatial parallelism by dividing the entire geographical region of a city into an array of blocks. Figure 3 illustrates the division of San Francisco downtown into nine blocks. Keeping our image size similar to MNIST (Salimans et al., 2016), each block is set to represent an image of size $24 \times 24$ pixels, with each pixel representing one $50m \times 50m$ square area. Hence, each block covers an area of $1200m \times 1200m$.

---

[3]One could easily extend this approach to a label within the set $\{0, 1, ..., 287\}$ if looking at labels associated with any five minute slots of a day or the set $\{0, 1, ..., 2015\}$ if looking at labels associated with any five minutes slots of a week.

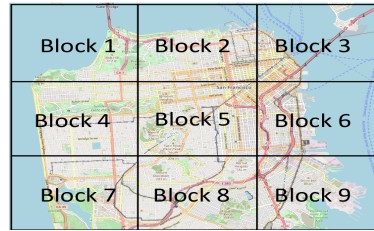

Figure 3: An illustration of how the geographical region of each city is divided into smaller blocks of equal size and trained independently. Note: image above is not drawn to scale.

Each block, representing a grey scale image of $24 \times 24$ pixels, depicts all the ride-locations in that block. Separate images are formed for pickup and drop-off locations; models trained are also separate for pickup and drop-off locations. Each image of a block is labeled with a time interval (for our experiments, the hour in a day) which is similar for both images created from pickup and drop-off locations. The synthetically generated images from an array of blocks with the same time interval label are combined by stitching together all the processed blocks of a city.

The generator network takes an input of a 100-dimensional Gaussian noise sample as well as a one-hot vector encoding of the time snapshot to be generated. It has a single, fully-connected hidden layer without any convolution (Goodfellow, 2018) consisting of 128 ReLU-activated neurons which then passes to a sigmoid activated output layer with the same number of output neurons as the total number of pixels in each block.

The discriminator network has a single hidden layer of 128 ReLU-activated neurons with a single sigmoid activated output neuron. We find that small networks are appropriate for the training data and allow for a quick and stable convergence to be achieved between the discriminator and the generator. Using relatively simple network architectures makes it possible to ensure that the discriminator and generator are *evenly matched* such that the loss for either network does not saturate early in the training process.

In addition to the standard GANs architecture of generator and discriminator, an additional network is introduced which is referred to as the classifier (Lee & Seok, 2017); it is pre-trained on the training data with the five minute label of the data serving as the classification target. In this way the time information that is encoded into the synthetic data by the generator network is then decoded by the classifier network. The generator is then trained on a weighted sum of the loss from both the classifier and discriminator networks as shown in the following equation:

$$\beta \log D(G(z,y)) + (1-\beta) \log C(G(z,y)) \tag{4}$$

where $\beta$ is a tune-able hyper-parameter.

This allows for more explicit loss attribution such that the generator receives two different error signals; one indicating the realism of the synthetic data and the other indicating accuracy relative to the conditioning values. By experiments using MNIST data and (Lee & Seok, 2017), we found adding a classifier increases the efficiency of the training process and results in higher quality synthetic data while incurring considerably less training time than other conditional GANs architectures we have experimented.

## 4 EXPERIMENTAL RESULTS

In this section, we present the cloud infrastructure used for running our experiments. We also present performance results on scaling our GANs workloads on the cloud infrastructure.

### 4.1 RUNNING GANS ON AWS

All experiments are conducted on Amazon Web Services (AWS) using c5.18x instances with each instance containing an Intel Xeon Scalable Processor with 72 virtual cores (vCores) running at 3.0GHz and 144 GB of RAM. In this work we set the block size for each of the four cities to be

| City | Number of Blocks |
|------|------------------|
| San Francisco | 1402 |
| Los Angeles | 1978 |
| New York | 765 |
| Chicago | 1155 |

Table 1: Training Workload for Different Cities. Each block is trained independently. Doubling the value provided in the table for each city, would give the approximate number of models trained because we have pickup and drop-off locations trained and generated separately.

$1200 \times 1200$ meters; each block is trained separately. Enlarging the block size will increase the computational time for training; and the complexity of the model can potentially impact scalability. The total number of blocks for each city are shown in Table 1. The number of blocks are mainly determined by the size of the greater metropolitan area of each city.

To help enhance the scalability of our GANs workload across multiple nodes we make use of Ray (Moritz et al., 2017) from Berkeley, a distributed framework for AI Applications, to efficiently parallelize our workload across cluster of CPU nodes on AWS. Ray provides a convenient API in Python to scale deep learning workloads for numerous libraries, and support for heterogeneous resources like CPUs and GPUs. We also make use of Intel's Math Kernel Library Intel (2018b) (MKL) which provides machine learning libraries for supporting operations like activation (ReLU), inner product, and other useful functions (Intel, 2018a).

## 4.2 TRAINING TIME

Using Ray we scale our training runs by using from 2 to 8 c5.18x instances (containing from 144 cores to 576 cores) on AWS. The scalability results are shown in Figure 4. As can be seen increasing the number of c5.18X Xeon CPU instances can significantly reduce the GANs training time up to 8 c5.18x instances. For the city of Los Angeles, the training time can be reduced from over one hour to less than 20 minutes. For New York City the training time can be reduced to just minutes. Running times for sampling ride requests from the trained models and stitching the images of all the blocks together are significantly less than the training times, and are not included in these results.

We also conduct our GANs scaling experiments using GPU instances on AWS. In our initial experiments we observe no real performance improvements using GPUs. Training time using GPUs on AWS was observed to be $5.93$ hours on a p3.8xlarge instance using NVIDIA's Multi-Process Service (MPS) (Nvidia, 2018). With MPS, the GPU utilization is close to maximum by running multiple of our small GANs training jobs in parallel on a single GPU. Although, the number of jobs which could be executed in parallel on a GPU are not that many in comparison to Xeons. Scaling on GPUs requires more investigation. In this work, we show that it is possible to achieve very nice scalability of our GANs workload using only CPU cores supported by Intel's MKL library and Berkeley's Ray framework.

## 5 VALIDATION OF SYNTHETIC DATA SETS

### 5.1 SPATIAL VARIATION

The correlation fractal dimension ($D_2$) gives a bound on the number of ride requests within a geographical region. This is an essential characteristic to match for the data set we are generating using GANs. In Tables 2 & 3, we provide the fractal range for each city within which the fractal dimension remains constant. It is important to note that the fractal range for each city differs. The fractal range provides the $\epsilon$ range for which the data exhibits statistical self-similarity (Belussi & Faloutsos, 1998). The variation in the fractal ranges for the different cities can be attributed to the geographical shape of the city for which the ride requests are generated. We hypothesize that due to Los Angeles's sprawling nature, a larger $\epsilon$ is needed to observe self-similar patterns in comparison to the other three cities, which have a more corridor-like geographical region.

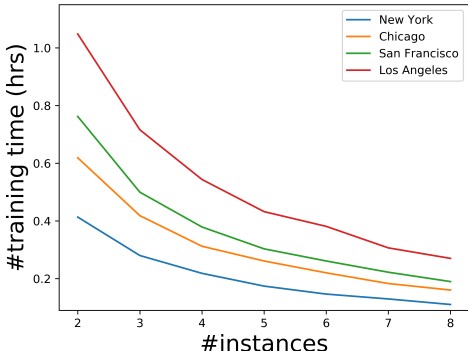

Figure 4: Training Time Performance Results for training our GANs for pickup locations on AWS with c5.18xlarge Xeon CPU instances. Results for training using drop-off locations show a similar trend.

| City | Fractal Range | Real Data Sets | | | Synthetic Data Sets | | |
|---|---|---|---|---|---|---|---|
| | | $D_2$ min. | $D_2$ max. | $D_2$ mean | $D_2$ min. | $D_2$ max. | $D_2$ mean |
| New York | (450, 2500) | 1.441 | 1.753 | 1.647 | 1.415 | 1.663 | 1.541 |
| Chicago | (600, 3000) | 1.139 | 1.558 | 1.384 | 1.188 | 1.567 | 1.435 |
| San Francisco | (450, 2500) | 1.283 | 1.731 | 1.548 | 1.242 | 1.65 | 1.426 |
| Los Angeles | (1500, 4000) | 0.962 | 1.638 | 1.355 | 1.048 | 1.489 | 1.314 |

Table 2: Summary of measured correlation fractal dimensions ($D_2$) for four cities; computed over a day for every hour using **pickup** locations of real and synthetic data sets.

One may also interpret $D_2$ as a way to measure the fidelity of generated images to that from real data. Comparison of the ranges of values of $D_2$, in terms of min, max, and mean values, for the real and the synthetic data sets are fairly close although not identical. In most instances the mean value for $D_2$ is lower for the synthetic data sets in comparison to the real data sets. We believe this discrepancy in the values of $D_2$ require further investigation. Recent works to improve capture learning of high-resolution details of an image (Karras et al., 2017) can potentially benefit the learning for our ride request images.

## 5.2 TEMPORAL VARIATION

DPL provides a characterization of the temporal evolution of ride requests. In the top row of Figure 5 we observe the plot of the DPL exponents $\alpha$ (slop of the line) based on the temporal patterns of the real data sets. For the ride request graph to obey DPL properties, we use graph generator proposed by (Jauhri et al., 2017a) to connect source and destination locations. In the bottom row of Figure 5 we see the same results based on the synthetic data sets. We can see that the DPL exponent values $\alpha$ correlated quite nicely with that from the real data sets for New York, Chicago, and San Francisco.

| City | Fractal Range | Real Data Sets | | | Synthetic Data Sets | | |
|---|---|---|---|---|---|---|---|
| | | $D_2$ min. | $D_2$ max. | $D_2$ mean | $D_2$ min. | $D_2$ max. | $D_2$ mean |
| New York | (450, 2500) | 0.613 | 1.718 | 1.498 | 0.914 | 1.595 | 1.414 |
| Chicago | (600, 3000) | 0.315 | 1.47 | 1.216 | 0.378 | 1.539 | 1.234 |
| San Francisco | (450, 2500) | 0.499 | 1.689 | 1.363 | 0.570 | 1.685 | 1.335 |
| Los Angeles | (1500, 4000) | 0.123 | 1.487 | 1.036 | 0.214 | 1.456 | 1.042 |

Table 3: Summary of measured correlation fractal dimensions ($D_2$) for four cities; computed over a day for every hour using **drop-off** locations of real and synthetic data sets.

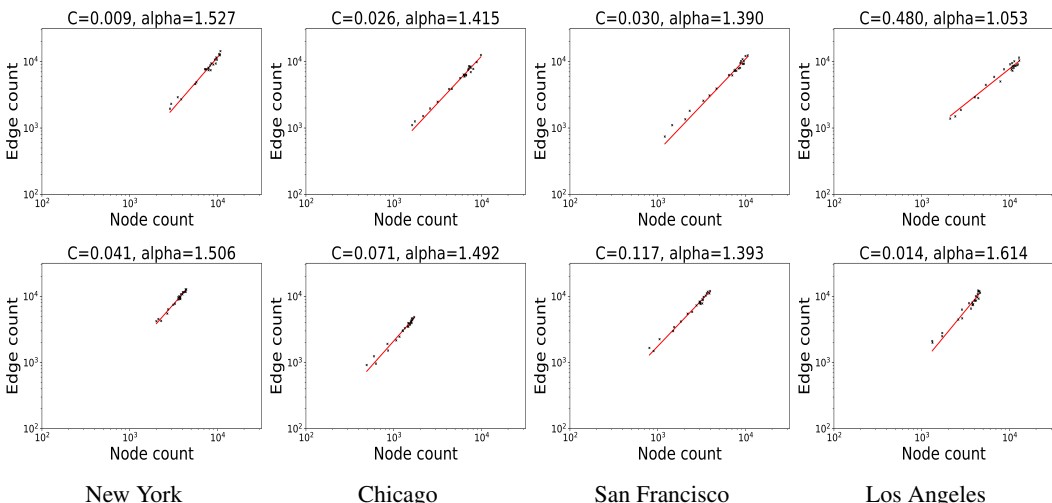

Figure 5: DPL plots from real data (top row) and synthetic data (bottom row) for four cities. The red line is the least square fit of the form $y = Cx^\alpha$, where $y$ and $x$ are number of edges and nodes respectively. $R^2 \approx 1.00$ for all of them.

For Los Angeles, the synthetic exponent is higher than the real observed value; the geographical region for LA is much larger and due to many prominent regions of high request density, the model may likely suffer from bias towards generating more requests in prominent regions leading to a faster increase of the number of edges connecting nodes present in high density regions.

Another validation of our GANs approach is provided in Figure 6. Here we observe temporal variation of ride requests in terms of the volume of ride requests generated for each hour of a typical weekday. We see that for all four cities, the temporal variation of the synthetic data sets match quite well the temporal variation exhibited by the actual data set.

## 6 CONCLUSION

The emergence of ride sharing services and the availability of extensive data sets from such services are creating unprecedented opportunities for: 1) doing city-scale data analytics on urban transportation for supporting Intelligent Transportation Systems (ITS); 2) improving the efficiency of ride sharing services; 3) facilitating real-time traffic congestion prediction; and 4) providing new public services for societal benefit. Moreover, the power of neural networks for machine learning has allowed the creation of useful models which can capture human behavior and dynamic real-world scenarios. The key contributions of this paper include:

- We map the ride requests of ride sharing services into a time sequence of images that capture both the temporal and spatial attributes of ride request patterns for a city.

- Based on extensive real world ride request data, we introduce a GANs based workflow for modeling and generating synthetic and realistic ride request data sets for a city.

- We further show that our GANs workload can be effectively scaled using Xeon CPU clusters on AWS, in reducing training times from hours to minutes for each city.

- Using previous work on modelling urban mobility patterns, we validate our GANs generated data sets for ride requests for four major US cities, by comparing the spatial and temporal properties of the GANs generated data sets against that of the real data sets.

There are other promising avenues for further research. Some open research topics include:

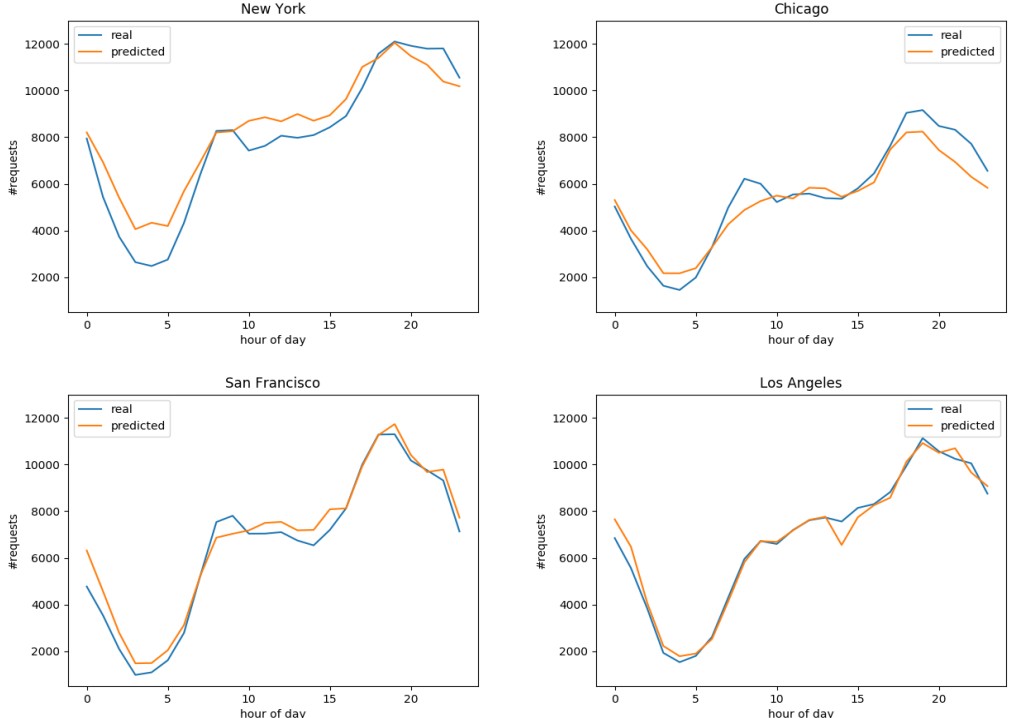

Figure 6: Plots for four cities highlighting the temporal variability of ride requests visible in both real and our model (predicted) for ride request generation. The pattern is representative of any typical day of week.

- Using the GANs generated data sets for experiments on new algorithms for dynamic ride pooling, real-time pre-placement of vehicles, and real-time traffic congestion prediction.[4]
- Using the GANs generated data sets for conducting experiments on *what-if* scenarios related to traffic congestion prediction and mitigation, and planning for future development of transportation infrastructures.

We are currently pursuing these research topics. As our GANs generated data sets are used in our follow up research, we plan to further validate the synthetic data sets by comparing our research results with results from using the real data sets. We plan to continue to tune our GANs models and generate improved synthetic data sets that can be made available for other researchers.

---

[4]We plan to publically release the code and models of our GANs along with algorithms for looking at transportation related problems like pooling and placement.

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
