# OpenReview forum: "Using GANs for Generation of Realistic City-Scale Ride Sharing/Hailing Data Sets"
_ICLR.cc/2019/Conference_

### Official Review · AnonReviewer2 · 2018-11-01
**Application paper of GAN based on known techniques**

**Rating:** 5
**Confidence:** 4

**Review:**

The paper produces a heat-map of ride-share requests in four cities in the USA. For each city 'block' they produce a time-sequence of 2016 images representing a week-long run from combining each 5-minute interval. This is used with a GAN to produce new data. The techniques applied, although not commonly used in the context of ride sharing / hailing, have been used extensively in other literature.

Some major points on the paper:
1) A GAN approach is normally used to generate more data when enough real data is not obtainable. However, here you only use one week of data from a much larger set. Surely, it would be better to make use of all the weeks available?

2) It is not clear how the heat-maps once produced could be used in the future. There is a hint in the results section about how they can be converted back to ride requests, but this is not clearly defined.

3) There are a number of cases where you state that some approach has been found to be better. However, no evidence is presented for how you determined this to be true.

4) The conversion of data to heat-maps has been used extensively in prior research. Although I'm not directly aware of the use in machine learning I am aware of the use in transport - "Interactive, graphical processing unit- based evaluation of evacuation scenarios at the state scale". The novelty here seems to be the application to this specific problem.

More specific points:
- "Our real ride request data sets consist of all the ride requests for an entire week for the four cities." - it's not clear - are all four cities used to train one model?

- "Hence the week-long data should be quite representative." - This fails to take into account such things as national holidays or other major events such as sports. Did your chosen week contain one of these?

- "Hence we believe the ride request data sets also reflect the overall urban mobility patterns for these cities." - This is a huge assumption, which would seem to need evidence to back it up.

- "and lump together all the ride requests within each interval." - Presumably you mean that all time values are to the granularity of 5 minutes?

- "We arbitrarily sized each block to represent an image of 2424 pixels" - this seems particularly small.

- "Each image of that block is labeled with a time interval (for our experiments, the hour in a day)." - Can the variability within an hour not make this more difficult?

- "We find that small networks are appropriate for the training data" - evidence to support this.

- "This network is pre-trained on the training data" - which training data are you referring to?

- "This is found to increase the efficiency of the training process" - evidence?

- "In this work we set the block size for each of the four cities to be
1200  1200 meters" - how was this value arrived at?

- You state that GPUs were no more efficient, it would be good to see more analysis of this.

- "To help enhancing the scalability" -> "To help enhance the scalability"

- "and other useful functions" - such as?

- Figure 4 would probably work better as a speedup graph.

- "Running times for sampling ride requests from the trained models and stitching the images of all the blocks together are significantly less than the training times, and are not included in these results." - at least some figures to give an idea of scale should be provided.

---

> ### Author Response · Authors · 2018-11-27
> **Rebuttal**
>
> [Thanks a lot for your valuable feedback and time.]
> - A GAN approach is normally ... all the weeks available?
> >>> Yes, using data from multiple weeks would have been better. However our initial analysis of the data, there is a very strong correlation of the ride request patterns from week to week for each city. We made the decision to use one week of real data for this paper since the actual amount of ride request data for each city is already quite sizeable. Future work can certainly examine data from multiple weeks.
>
> - It is not clear how the heat-maps ... this is not clearly defined.
> >>> We have added text to the first paragraph under section 2.1. Essentially, after generating locations (pickup and drop-off) using GANs, we use graph generators to pair GANs generated pickup locations and drop-off locations to obtain synthetic data sets with complete ride request information.
>
> - The conversion of data to ... to this specific problem.
> >>> Also, metrics (DPL and fractal dimension) to understand such a dynamic phenomena has not been studied before. (see responses to Reviewers 1 and 2)
>
> - "Our real ride request data sets ... one model?
> >>> We use the real data from each city to train the model for that city. From previous publications, we know that each city can exhibit different ride request patterns. Hence we adopted the city-specific approach. For each city, we do partition the city into blocks and perform the training process for the blocks in parallel and then stitched the end results from all the blocks of a city into one data set for that city.
>
> - "Hence the week-long data should be quite representative." ... one of these?
> >>> We have looked at weeks with holidays, rainy days, and other anomalies. Such events have an impact on the ride request pattern. Our goal in this work is to highlight how training GANs using a representative week of data can be used to generate realistic ride requests that capture the temporal and spatial properties for a typical week. To be able to account for (and potentially predict) specific anomalies, such as holidays, weather condition, special events, etc., is a very interesting topic for future research.
>
> - "Hence we believe the ride request ... evidence to back it up.
> >>> Yes, this belief is based on our optimistic assumption. We have carefully selected the four US cities where ride sharing/hailing services have very high penetration rates. Also these penetration rates have continued to increase in these cities. Based on previously published papers in related areas such as traffic flow modelling, the necessary sampling rate for sampled data set to be representative for overall traffic flow is usually lower than the penetration rates of ride sharing services in these cities.
>
> - "and lump together all the ... of 5 minutes?
> >>> Yes.
>
> - "Each image of that block ... more difficult?
> >>> Variability  does make it difficult but due to paucity of data from multiple weeks we went with this approach. Although, in the footnote on page 4 we do highlight that the labels can easily be modified if data from multiple weeks is available. (see response to other reviewers)
>
> - "We find that small networks ... to support this.
> >>> We did experiment with increasing the number of hidden layers for MNIST and our data set. In both cases, the difference in the images generated was negligible. Also, work on a better network with potentially convolutional layers and/or RNNs is planned for future.
>
> - "This network is ... are you referring to?
> >>> Clarified in the revised paper. We are referring to the images with pixel values as the ridership and label as the time snapshot at which the image is captured.
>
> - "This is found to ... - evidence?
> >>> By experiments using MNIST and ride request data; and prior work by Lee, “Controllable generative adversarial network.”
>
> - "In this work we set ... value arrived at?
> >>> We wanted to keep a simple model with image size similar to MNIST which has been studied extensively using GANs. (see response to Reviewer 2)
>
> - You state that GPUs ... analysis of this.
> >>> We added our best result for training on GPUs to the revised version. Frankly the experimental results using GPUs did not give us better performance. We suspect that in the AWS infrastructure the efficiency of using large clusters of GPUs is not as scalable as clusters of CPUs, especially enhanced by Intel’s MKL and Berkeley’s Ray. It is also possible that the virtualization of GPUs comes with a higher overhead than that for the CPUs. This thread of work specifically for our application and other similar ones require more investigation.
>
> - "and other useful functions" - such as?
> >>> Added reference to Intel’s MKL primitives.
>
> - "Running times for sampling  ... idea of scale should be provided.
> >>> Sampling and stitching images takes a couple of minutes for any city on a laptop with a reasonable hardware configuration.

---

> > ### Comment · AnonReviewer2 · 2018-12-05
> > **Thanks for the clarification**
> >
> > Thanks for clearing up some of the issues in the paper. I appreciate that you want to keep this paper 'simple' in terms of model and do other work in the future. However, I feel that you would need to do some of these other items to get this paper to a higher level for a top conference such as this one.

---

### Official Review · AnonReviewer3 · 2018-11-02
**An interesting application of GAN**

**Rating:** 5
**Confidence:** 4

**Review:**

The paper works on a very interesting problem: generating ride hailing demand map using deep learning technique. The idea is novel and interesting. The paper adopts two metric to evaluate the performance of the algorithm and shows that the performance is good. However, the problem is a little far from real world cases, which limits the contribution of the paper.

The title of the paper is very attractive. Before reading the paper, I was very excited and wanted to see the algorithm could generate the driving trajectory by using GAN. However, it can only generate the pickup location, which makes me a little disappointed. In real world applications, in riding hailing industry, the demand/supply estimation have been wide investigated. And it can be very accurate. It is not clear why we need to generate it. On the other hand, the paper only adopts conventional GAN to this application. Technically the contribution is not significant. The paper only considers time slot for generating the new data. In this area, much more information has been used. The authors are suggested to survey the smart transportation or riding sharing research area. The training solution is not satisfied. The model is only trained for each small area in the city. The training set is very limited, which may make the model overfit. Thus the experiments and the results are not convincing. In current GIS or transportation area, usually we would use a unique model for the whole city. At least the authors should discuss their algorithm for the scale issue.

Overall, the paper works on a very interesting problem. However, the current solution should be improved.

---

> ### Author Response · Authors · 2018-11-27
> **Rebuttal**
>
> [Thanks a lot for your valuable feedback and time.]
>
> - However, it can only generate the pickup location, which makes me a little disappointed.
> >>> This has now been clarified in the paper. We did generate pickup points and drop-off points using separate GANs models. (see our response to Reviewer 1)
>
> - In real world applications, in riding hailing industry, the demand/supply estimation have been wide investigated.
> >>> Having generated both pickup and drop-off points, we can generate an entire ride request (pickup location and time + drop-off location and time) as opposed to just predict how many requests are going to be generated from a particular area. This type of fine-grained (both temporally and spatially) generation of ride request has not been found in prior literature. We do this at very fine granular time intervals (5 min) and spatial areas (50mx50m) for very large geographical regions covering entire greater metro area of a large city.
>
> - Technically the contribution is not significant.
> >>> Converting ride requests into images is a novel idea. Importantly, our application provides a very relevant and real-world application of GANs. Also, metrics (DPL and fractal dimension) to understand such dynamic phenomena like real-time on-demand ride requests have not been studied before.
>
> - The model is only trained for each small area in the city. The training set is very limited, which may make the model overfit.
> >>> Our model is actually trained for the entire city and we use training data set for the entire city. The reason for partitioning the entire city into blocks that are trained separately is to exploit  parallelism available in the AWS training infrastructure so as to reduce the overall training time. The final GANs generated data set covers the entire city by stitching together results from all the blocks of a city. The actual data set in terms of the total number of ride requests is quite sizable and much larger than most other research data sets.
>
> - Thus the experiments and the results are not convincing.
> >>> We have added additional data in the revised version of the paper highlighting close similarity, based on the fractal dimension metric, for both the pickup and dropoff points. (see our response to Reviewer 1)
>
> - In current GIS or transportation area, usually we would use a unique model for the whole city. At least the authors should discuss their algorithm for the scale issue.
> >>> Ideally we want to use one single model for the entire city, i.e. not partition into blocks. However, this will make the total training time unacceptable because of the inability to leverage parallelism of the training infrastructure. By partitioning the entire city into blocks, we create spatial parallelism that can take advantage of infrastructure parallelism to reduce the training time. Yes, by increasing the block size we can potentially capture larger patterns that span multiple blocks. However, this can significantly increase training time. So, it is a trade off. Based on our experimental results, our selected block size of 1200mx1200m seems to perform quite well in capturing the ride request patterns for SF, NY, & Chicago. For these three cities the fractal range that captures the spatial distribution patterns all start at much lower values. (see Table 2 and Table 3). Increasing the block size for these cities is not necessary. Only LA has a fractal range starting at a much higher value, with the lowest mean fractal dimensionality. For LA, increasing the block size can potentially improve these results by capturing larger (spatially) patterns.

---

### Official Review · AnonReviewer1 · 2018-11-05
**Interesting idea, weak eval**

**Rating:** 4
**Confidence:** 4

**Review:**

The authors propose an interesting idea of generating synthetic data sets for ride sharing. In particular, they split the space/time into small spatial/temporal cells (50mx50m and 5min) each containing number of requests (or a scaled version of it), and train a conditional GAN to output these cell values given an input 5-min time label. They validate the results using metrics from graph and fractals theory.
While the idea is interesting, the execution of the paper is lacking. Some details are missing and especially key things such as metrics should be explained better.
- How is the data represented? It says that pixel represents the number of ride requests, how exactly? Then, in the next paragraph it is said that pixel represents presence/absence of ride requests, so which one is it. This is a critical part of the proposal and is not well explained.
- y-axis in Figure 2 is not explained.
- Metrics should be better explained. How are edges defined, when you only model requests, not destinations? This is far from being clear.
- In addition, how is D2 defined? Do we compute one for each time, or how? What exactly is "side e", what is a "side" here? Basically both metrics are not well defined.
- "We can claim strong similarity ...", what is this justified by?
- Second paragraph in Section 3.1 is not clear, reads very strangely.
-  Labels are being mentioned before being defined, adding to confusion.
- It is clear from Figure 2 that workdays and weekends are very different, yet the authors chose to ignore that fact during modeling. They do mention that we can choose any labeling we want, but still strange that for the experiments this was not taken into account.
- The authors mention that cells as 1.2km x 1.2km, but Figure 3 shows much different resolution. Seems that the figure is just given as an example, but reading the text one gets an impression that the figure was actually used in the paper. This needs to be clarified.
- For the classifier, it says that "time sequence of the data" is a label, what does this mean? You mean the actual label, or some time sequence? This is confusing, although it seems that simply the 5-min label was used.
- Could we add the metrics to the loss, to enforce them as the authors say that that would result in strong similarity?
- One of the major flaws of the paper is missing baseline. It is very difficult to appreciate the results without any reference result.
- Again, I am not sure how results in Section 5.2 are computed when only requests are modeled.
- Footnote 2 in the conclusion mentions baselines, yet there are none mentioned in the paper.

---

> ### Author Response · Authors · 2018-11-27
> **Rebuttal**
>
> [Thanks a lot for your valuable feedback and time.]
> - How is the data represented? It says that pixel represents the number of ride requests ... is not well explained.
> >>> Pixel intensity is set to zero if there is no ride request in that 50mx50m square and to positive integers when present with values reflecting the number of ride requests in that 50mx50m square (in the time interval captured by that image). This has been clarified in Section 2.1 of the revised version.
>
> - y-axis in Figure 2 is not explained.
> >>> y-axis is the total ridership in a city (for each time interval).
>
> - Metrics should be better explained. How are edges defined, when you only model requests, not destinations?
> >>> Each edge is directed from source (pickup) node to destination (drop-off) node. Edge weight represents the number of requests originating and ending from the same pickup to destination nodes (a node represents a 50mx50m square). More details from prior work about the construction of the ride request graph have been added to section 2.2.
>
>  - In addition, how is D2 defined? Do we compute one for each time, or how? What exactly is "side e", what is a "side" here?
> >>> Yes, D2 is computed for each time snapshot. Side e (\epsilon) is the side length of the bounding square area, e.g 50mx50m square. Footnotes (on page 3) and related sections have been modified.
>
>  - "We can claim strong similarity ...", what is this justified by?
> >>> It is justified in figure 2 in terms of temporal fluctuation of total ridership, and also by the two summary metrics -- DPL exponent, and D2 which we leverage from previous publications. For each week the deviation in these metrics is negligible which highlights a pattern. In general, human commuting properties if quantified by metrics wouldn’t differ a lot since people residing in urban cities tend to have a consistent pattern like visiting offices in the morning; visiting restaurants and bars in the evening. Anomalies are bound to happen but on average such anomalies, generally short lived in time and constrained in space, will not have much impact on the overall average of the two metrics we are computing. Essentially these two metrics serve as  very useful first order characterizations of temporal and spatial variations in the mobility patterns within a city.
>
> - Second paragraph in Section 3.1 is not clear, reads very strangely.
> >>> Corrected in revised version.
>
> - It is clear from Figure 2 ... chose to ignore that fact during modeling.
> >>> This is our initial attempt at this problem. We train our models using data from weekdays and weekends without separating them. If we had access to data from multiple weeks, we would have trained weekdays and weekends separately leading to much better generators. Actually we can go even further to train a different model for each day of the week based on potential variations even within weekend and weekday groups.
>
> - The authors mention that cells as 1.2km ... resolution.
> >>> The figure is not drawn to scale is only for illustration; this is now clarified in the paper.
>
> - For the classifier, it says that "time sequence of the data" is a label, what does this mean? ... simply the 5-min label was used.
> >>> Yes, it is simply the label for each 5-min interval.
>
> - Could we add the metrics to the loss, to enforce them as the authors say that that would result in strong similarity?
> >>> Not sure how helpful that would be. We would prefer the GANs to understand the pattern of requests without explicitly highlighting the metrics specific to the domain which capture the dynamism of the phenomena.
>
> - One of the major flaws of the paper is missing baseline.
> >>> To the best of our knowledge, prior work has focused on demand and supply generation i.e. the number of people who will request at a particular location or the number of vehicles which will be required at some geographical location. We are generating the entire ride request (source to destination) and then proposing metrics which capture the spatial and temporal variations of such ride requests. This paper focuses on a novel way to generate synthetic data set and the validation of the data set using the two metrics for temporal and spatial attributes. The relevant comparison with any baseline would need to involve implementing some application on top of and using the two different data sets (real and synthetic). The results from the real data set then become the baseline. We are pursuing a number of such applications.
>
> - Again, I am not sure how results in Section 5.2 ... requests are modeled.
> >>> This has been corrected. We did study both pickup points and drop-off points using separate GANs models. In the initial submitted version we only show the data for the pickup points. We have added D2 metric table to highlight the similarity between real and synthetic datasets for both pickup and drop-off locations.
>
> - Footnote 2 in the conclusion ... in the paper.
> >>> Clarified in the revised paper.

---

### Meta-Review · Area_Chair1 · 2018-12-11
**Interesting application of GANs but weak evaluations**

**Confidence:** 5
**Recommendation:** Reject

**Metareview:**

While the reviewers all agree that this paper proposes an interesting application of GANs, they would like to see clearer explanations of the technical details, more convincing evaluations, and better justifications of the assumptions and practical values of the proposed algorithms.